# Perspectives of Health Care Providers on the Role of Culture in the Self-Care of Patients with Chronic Heart Failure: A Qualitative Interview Study

**DOI:** 10.3390/ijerph17145051

**Published:** 2020-07-14

**Authors:** Adam Jönsson, Emilie Cewers, Tuvia Ben Gal, Jean Marc Weinstein, Anna Strömberg, Tiny Jaarsma

**Affiliations:** 1Faculty of Medicine and Health Sciences, Linköping University, 58183 Linköping, Sweden; adam.m.joensson@gmail.com (A.J.); emilie.cewers@hotmail.com (E.C.); 2Heart Failure Unit, Cardiology Department, Rabin Medical Center, Petah Tikva 49100, Israel; bengaltu@gmail.com; 3Sackler Faculty of Medicine, Tel Aviv University, Tel Aviv 6997801, Israel; 4Cardiology Division, Soroka University Medical Centre, and Faculty of Health Sciences, Ben Gurion University of the Negev, Beer-Sheva 84105, Israel; jmwein@bgu.ac.il; 5Department of Health, Medicine and Caring Sciences, Linköping University, 58183 Linköping, Sweden; anna.stromberg@liu.se; 6Department of Cardiology, Linköping University Hospital, 58183 Linköping, Sweden

**Keywords:** self-care, intervention, culture, qualitative health care providers

## Abstract

Background: Self-care is important in chronic diseases such as heart failure. The cultural background of health care providers might influence their view on self-care behaviour and education they provide. The aim of this study was to describe health care providers’ perceptions of the role of culture in self-care and how those perceptions shape their experiences and their practices. Methods: A qualitative study was performed in Israel, a country with a culturally diverse population. Data were collected using semi-structured interviews with 12 healthcare providers from different cultural backgrounds. Interviews were recorded and transcribed verbatim and analysed using content analysis. Results: Healthcare providers experienced cultural background influenced their patients’ self-care behaviour. Perceived culture-specific barriers to self-care such as dietary traditions interfering with the recommended diet, willingness to undertake self-care and beliefs conflicting with medical treatment were identified. Healthcare providers described that they adapted patient education and care based on the cultural background of the patients. Shared cultural background, awareness and knowledge of differences were described as positively influencing self-care education, while cultural differences could complicate this process. Conclusions: Cultural-specific barriers for self-care were perceived by health care providers and they identified that their own cultural background shapes their experiences and their practices.

## 1. Background

Optimization of self-care is a focus of healthcare worldwide in order to improve outcomes of patients with chronic diseases such as decreasing symptoms, improving well-being and even survival [1,2]. Self-care is a process of maintaining health through health-promoting practices, symptom monitoring and managing symptoms when they occur and is important in patients with chronic disease [1,3]. Several factors are known to be related to self-care, such as experience and skills, motivation, habits, cultural beliefs and values, functional and cognitive abilities, support, and access to care [4,5]. Self-care is strongly influenced by confidence, illness perception and self-efficacy [5,6,7]. Furthermore, cultural beliefs and values affect self-care behaviour, for example if a patient needs to prioritize one self-care behaviour over the other due to cultural habits or when the advice itself contradicts cultural beliefs.

Culture can be defined as ‘an internalized and shared schema or framework that is used by a (sub) group as a refracted lens to “see” reality and in which both the individual and the collective experience the world’ [8]. Culture may, for example, include people of the same religion or origin. Culture influences attitudes and behaviours in relation to lifestyle, and activities to promote health, which may subsequently lead to differences in self-care performance [9].

One of the chronic diseases in which self-care behaviour has been shown to be part of the comprehensive management is heart failure (HF). Self-care education is the focus of HF management programmes worldwide [10,11]. HF-specific self-care includes timely medication taking, maintaining a healthy diet, monitoring symptoms, quitting smoking and limiting the intake of alcohol [10,12]. Due to the significant benefits of self-care in improving outcomes for patients with HF, the role of healthcare providers in educating patients to perform self-care is vital. In the HF guidelines of the European Society of Cardiology, specific key topics and self-care skills to be included in patients’ education are highlighted and professional behaviours to optimise learning and facilitate shared decision-making are described [10].

Communication between patients and health care professionals is of vital importance. How a health care provider responds to a patient during a consultation or clinical interview influences how much information the patients will obtain, with certain groups of patients being more vulnerable to miscommunication [13,14].

Previous research emphasises that certain cultural groups regard healthcare providers as having great authority, which may increase patient adherence to self-care [15]. Dietary habits and the general lifestyle of patients may differ depending on their cultural backgrounds, which might make it difficult to perform self-care. Specific self-care behaviours might be perceived as more difficult in some cultures, for example patients from Eastern/Asian countries were found to experience greater difficulty adhering to sodium intake recommendations than patients from other countries [2,16]. Differences between countries and cultures might be based on different self-care guidelines depending on the country of residence, but it might also be because certain foods that belong to a specific cultures are often rich in sodium [2].

Healthcare providers present information and support skill building in patients on how to perform self-care but also need to address those cultural habits or different lifestyles that may make it difficult to follow the self-care recommendations [17,18].

At the same time, it is important to consider that the clinician is not “culture-free” and that the cultural background of those who provide the patient education can play an important role. Cultural competence is the ongoing process in which the health care provider continuously strives to achieve the ability to effectively work within the cultural context of the client (individual, family, and community). This ongoing process involves the integration of cultural awareness, knowledge, skill, encounters, and desire [19]. Cultural competence is also described as the professionals’ understanding of how culture affects their views and activities as well as the interventions that they apply [20]. Cultural competence can be addressed in basic education but it needs more than a course in basic training of health care professionals and should be considered throughout their entire career [21]. Cultural awareness is an important aspect of cultural competence and involves the recognition of one’s biases, prejudices, and assumptions about individuals who are different [19,22].

Although several studies have been published on the effect of improvement of cultural competence of students from professional health care educations internationally [23,24,25], little is known about the actual thoughts and assumptions of health care professionals working in daily practice with patients, related to self-care. There are several quantitative instruments [19,22,23] that measure general culture awareness of a person or a group; however, the way culture is perceived by a person to be related to self-care is scarcely studied. In-depth knowledge about their perceptions of the role of culture in self-care and how those perceptions shape their experiences and their practices can help us to understand the relationship between culture and self-care [4,26].

The aim of this study was to describe health care providers perceptions of the role of culture in self-care and how those perceptions shape their experiences and their practices. 

## 2. Methods

This study used a qualitative descriptive design. 

### 2.1. Setting, Participants and Sampling

#### 2.1.1. Setting

Interviews were conducted with healthcare providers working in the Heart Failure Outpatient Clinic in Rabin Medical Centre and in the cardiology department of Soroka University Hospital, both located in Israel. The sites were selected based on previous research collaboration. Both sites treated HF patients from different cultural backgrounds. Israel is a multicultural country due to mixing of immigrants from all around the world (the “diaspora”) back to Israel (Aliyah), and to the cultural diversity of the local inhabitants. This makes Israel an appropriate location for studying how culture influences self-care through the experiences of healthcare professionals from different backgrounds treating patients with diverse backgrounds. Rabin Medical Centre, situated near Tel Aviv, predominantly serves patients from the central and northern domains of Israel, meaning the more urbanised areas with relatively high socioeconomic status, more noticeably influenced by western culture. Soroka University Hospital, based in Be’er Sheva, serves patients from the southern domains of Israel, which are not as urbanised, not as influenced by western cultures and have a large Bedouin population. The Bedouin population historically consisted of Arabic nomads living in the deserts of what is now Israel, among other places. In modern times, there are not as many Bedouins living a nomadic lifestyle. The city of Be’er Sheva itself is a westernised town with a predominantly Jewish population. 

#### 2.1.2. Selecting the Sample

The goal with selecting the study sample was to find the greatest variation in cultural background possible; therefore, a convenience sample was used, the participants were chosen in collaboration with employees who were familiar with their cultural backgrounds. A key informant (Tuvia Ben Gal and Jean Marc Weinstein) in each centre suggested and introduced the interviewers to health care professionals with a variety in cultural background (mix of religious, ethnic background, duration of stay in the country, urban/non-urban living) and invited them to participate. Since the main goal was not to focus on professions, we did not strive for an equal balance in health care professions. 

#### 2.1.3. Participants

The professions of the participants included medical doctors, interns, nurses and physiotherapists. There was no relationship between the researchers and the participants before the study. The cultural background of the participants varied between Arabs from Israel, “sabras” (Jewish people born in Israel) and people who have made the “Aliyah” (people of Jewish heritage who have re-located to Israel from different countries) and one participant from a Bedouin tribe in Israel.

Details on the participants were included on a group level (see Table 1). In total, 12 participants were invited and interviewed in English, none refused. Some interviewees were fluent in English, since it was their first language, others were not as fluent, but they confirmed feeling comfortable sharing their thoughts in English. The participants were healthcare providers from different professions, including: one nurse, one physiotherapist, three residents in cardiology and six cardiologists.

The age of the participants varied greatly, some were recently qualified health care professionals and there were participants past retirement age, but still active health care professionals.

### 2.2. Interviews and Procedures

#### 2.2.1. Research Team

The research team was a blended group related to clinical and research experience and related to cultural background, with three men and three women, 3 members of the team being Swedish, 1 Dutch and 2 Israelis with ages ranging from 22 to 60 years of age. Two were living in Israel, 1 had lived in the West Bank, 2 spent three months in Israel during a research project. Three members were born in cultures other than the one they were currently living in, and four members had experience as medical health care professionals caring for patients from several cultures.

#### 2.2.2. Interviews

Semi-structured interviews were used to collect data. The interviewer used an interview guide to help direct him through the interview process, but left room for any thoughts that the participant wanted to share. It maintained some structure, but it also provided the researcher with the ability to probe the participant for additional details [27]. The interview guide that was used was created in collaboration between Adam Jönsson, Emilie Cewers, Tiny Jaarsma, Tuvia Ben Gal and Jean Marc Weinstein with open-ended questions and probes [27]. This guide was tested in a pilot interview, after which it was revised in order to better fit the aim of the study and make it more comprehensible to the participants (see Appendix A). This included adding relevant questions and changing phrasing. Data from the pilot interview was not included in the analysis. The questions asked were intended to gather information about participants’ backgrounds, their thoughts regarding the influence of cultural background on HF self-care, barriers faced by people from different cultural backgrounds and the influence of their own cultural background. Nine interviews were held at an office in the Rabin Medical Centre and three at the Soroka Hospital. The same individual (Adam Jönsson, a 3rd-year medical student with a Swedish cultural background) conducted all the interviews after introducing himself as described above. The interviews differed in length, the shortest being 22 min and the longest 54. Most were around 45 min long. After 11 interviews, no new information emerged, and a 12th interview was performed to ensure this: no new information was gathered. Another researcher (Emilie Cewers, a 3rd-year medical student with a Swedish cultural background) was also present during the interviews. Her role was to take notes during sections that were not recorded (these included questions not related to the aim of the study that were used as a warm-up for the participants and interviewer) and to support Adam Jönsson when needed during the interviews; for example, asking questions missed by the interviewer. 

#### 2.2.3. Transcripts

Interviews were recorded and transcribed verbatim by the interviewer. 

The original transcript was kept intact throughout the process. Summaries of the transcripts were made and sent to the interviewees, together with questions that arose during transcription. The interviewees were then asked to confirm, change or add any information deemed necessary. No additional information or comments were received. Finally, we sent the manuscript to three additional health care providers with different cultural backgrounds from the same setting to confirm the findings and check for harmful stereotypical wording or interpretation. As a consequence, referrals to very specific religious or cultural groups were deleted from the quotes; however, keeping the meaning of the quote intact.

### 2.3. Ethical Aspects

The Institutional Ethics Board of the Rabin Medical Center waived interview studies in health care personnel in general. We gained approval from the heads of department of both institutions and written informed consent was gathered from the healthcare providers before commencing the interviews and they gave approval to collect their demographic data. In the analysis and in the quotations provided, the different professions of the participants were not distinguished. To ensure confidentiality of the participants, the quotes do not reveal personal information, and the participants were coded with a number [28]. 

### 2.4. Analysis

The interviews were all completed before the analysis started. The data were analysed according to the qualitative content analysis method developed by Graneheim and Lundman [29]. The transcripts from all 12 interviews were each read through thoroughly one at a time in order to get an overview of the interview, after which sections concerning the same topic (meaning units) were extracted into a table, where they were condensed and then coded. There were 2 coders of the first interviews (Adam Jönsson and Tiny Jaarsma). The first two interviews were coded independently and compared, and consensus was reached after discussion. After this, one coder (Adam Jönsson) proceeded to code and Tiny Jaarsma checked the final codes. The codes from all 12 interviews were then formulated into a document and screened for similarities to be placed in distinct subcategories. No specific software other than Word and Excel was used.

Discussions with an experienced researcher (Tiny Jaarsma) took place during the coding process in order to increase reliability and also to get a different perspective on where the core content of the material was. A third researcher (Anna Strömberg) was involved in creating the subcategories and categories. Subcategories were revised, and the codes re-sorted to best fit the aim of the study. We created categories that identified and defined groups of subcategories and codes that shared common characteristics in order to compare them to other categories [30]. Quotes are added to illustrate and strengthen the findings. An example of the coding, subcategory and category can be found in Table 2.

### 2.5. Rigour

To assure confirmability, there were frequent discussions within the research team (Tiny Jaarsma, Adam Jönsson, Emilie Cewers), serving the purpose of using appropriate data collection techniques and analysis methods. The preunderstanding, beliefs and assumptions were also discussed, and testing of rival explanations was undertaken to avoid reporting on findings that were not fully supported by the original data. Credibility was established by using purposeful sampling, increasing the likelihood that healthcare providers from different cultures had the opportunity to provide input regarding the subject under investigation. This meant minimising the risk of extracting findings based only on opinions from one specific cultural group and providing a solid framework for analysis. The transcribed interviews were sent back to the participants to confirm content. The informants also received a draft of the paper in its current form for confirmation and the three additional health care providers with different cultural backgrounds from the same setting to confirm the findings. In terms of transferability, the data may be transferable to some extent, while some findings may only apply to the specific culture from which the data were collected [27].

## 3. Results

During analysis of the material, 124 codes were used, and these were subsumed in 11 subcategories and then into four categories. The four categories were ‘Culture permeates self-care behaviours’, ‘Culture influences the way care is provided’, ‘Mutual cultural background impacts the mindset to address self-care’ and ‘Culture is only a small piece of the puzzle…’ All four categories had a substantial number of codes but most of the codes were supporting the category that ‘Culture permeates self-care behaviours (see Table 3).

### 3.1. Culture Permeates Self-Care Behaviours 

The Health Care Professionals (HCPs) stated that the cultural background of the patients can influence their engagement in self-care in several ways. The role of culture may not only be important in relation to one specific self-care behaviour (e.g., exercise) but was experienced as being present in and permeating several self-care behaviours and even influencing several related factors (e.g., knowledge, attitude, finances). Culture, or belonging to a cultural group, was perceived to generate barriers to perform self-care, influence the willingness to adhere to self-care advice and to predispose a person to limited knowledge of health, acquired habits and closely defined gender roles. 

Participants elaborated on several barriers to specific self-care behaviours, for example smoking, dietary changes or seeking help in case of deterioration. The willingness to perform self-care was believed to differ among patients from different cultures both in relation to adhering to self-care maintenance behaviours and in utilising the available healthcare system. 

*…maybe that the people in our religion don’t take advantage of medicine, maybe they don’t go to the physicians…//… I think that it could be much more problematic in our religion*.*(HCP 8)*

Cultural background of patients was found to sometimes raise various challenges when it came to embracing self-care behaviours such as smoking cessation, diet restrictions, adherence to medical treatment and exercise. The health care providers perceived that some cultural groups might be more disciplined than others when it comes to their willingness to adhere to self-care advice.


*…I can tell you that this ethnical group takes care of their health very, very much you know, // Yeah, disciplined, so they do the physical activity, they take their medications…*
*(HCP 6)*

Some cultures might place specific value on certain types of self-care advice due to previous circumstances; for example, the value of fluid restriction might be different in the Bedouin population, where water is considered a source of life because of the shortage of water in the desert. 

*…people again, from the desert for example, that were, the water meant life and the, you tell them not to drink and suddenly clashes with the very essence of their lives […] so it comes even more difficult to convince those people not to drink*.*(HCP 3)*

Participants also described occasions where cultural norms impact patients’ ability to perform self-care; for example, situations where cultures considered it inappropriate for both genders to perform supervised physical activity in a mixed group or for women to perform physical activity alone or at all, which subsequently creates a barrier to performing self-care. 

Participants experienced differences in the level of health knowledge of patients from different cultural backgrounds which was considered a barrier to lifestyle changes. They shared experiences in which the religious beliefs of certain cultural groups superseded medical treatment, regardless of the specific religion involved. Superseding of medical treatment could originate from advice from religious leaders, from the belief that God will fix every problem or that every diagnosis is meant to be and therefore does not require treatment.

‘*They are believing in that, God will fix everything, or if it’s not fixed, this is what God meant for them*.’ *(HCP 4)*

Participants stated that the women in several cultures takes over the total (self-)care of their husbands when they get sick, indicating that the men rarely have to concern themselves with their own health. 


*Since the childhood the mentality is that the man is a king, and he does not have to do anything for himself//*
*(HCP 1)*

### 3.2. Culture Influences the Way Care is Provided

Several HCPs reported that they adapted their self-care education to the cultural backgrounds of patients. They adapt their language, their efforts or their approach. Some participants did not adjust recommendations according to cultural background, while others described having prejudicial thoughts when treating a patient from a certain culture or religion.

*I think it’s in the back of my mind all the time, like I told you, when I’m going to the ER and I see a new case of a very religious family, I understand that it might be more complicated than what I see*.*(HCP 4)*

Participants stated that adjusting care based on a patient’s cultural background is not necessarily negative. For example, knowledge about disease prevalence in certain cultures was described as important since certain comorbidities interfere with the active treatment of HF. Some barriers in adhering to self-care advice were more prominent in certain cultures, and it could therefore be beneficial to thoroughly explain the importance of the treatment accordingly. One HCP said:

*You should somehow adapt your instructions to whatever you think those people have at home. It is difficult because we don’t really live in those cultures, we try to imagine how it is in their places and try to talk to the patients and understand from whatever that brings up how they live, how their lives look like. And try to instruct them according to it*.*(HCP 3)*

Although beneficial at times, HCPs also mentioned that situations might arise where patients become offended if they feel as though they are being treated differently because of their cultural or religious background. One HCP said:


*People could get offended. I mean, if you tell… ‘You think I’m stupid?’ ‘I know how to take my medications! … I know that you think that in my culture people drink all the time, but I know my job and I’m not drinking that much.’*
*(HCP 3)*

### 3.3. Mutual Cultural Background Impacts the Mindset to Address Self-Care

The participants described benefits of a shared background since it increases the understanding between the patient and healthcare provider, and they perceived that this led to greater adherence to self-care advice. Participants explained that coming from the same cultural background might facilitate through speaking the same language and understanding or trusting each other more:

*It’s not only because [of the] language barrier, it’s just I think it’s in our nature to believe more to something same*.*(HCP 5)*

Disadvantages of different cultural backgrounds between patient and healthcare provider involved language barriers, uncertainties concerning cultural or religious norms and rules and more effort being required to explain medicine. Moreover, some HCPs mentioned that certain cultures or religions had difficulties accepting female professionals, which also creates a barrier in the relationship between patient and healthcare professionals. One HCP said:

*They think that the woman is better at home. And some did not accept me as a doctor; they felt more confident when there was a male doctor, and not a female doctor*.*(HCP 4)*

### 3.4. Culture is Only A Small Part of the Puzzle…

The participants described the importance of culture; however, at the same time they mentioned that there are more factors to consider when talking about self-care in HF patients and the influence of culture might change during the disease trajectory. Furthermore, some factors might be only partly explained by belonging to a particular cultural group. Many participants acknowledged religious differences related to attitude to self-care; however, others felt that it did not have anything do to with the religion. 

*…people from my religion, they take less medication than other people, but nothing that has to do with the religion*.*(HCP 8)*

Furthermore, differences within a cultural group were not only found to be related to certain beliefs in that group but were also due to other factors connected to the group, for example, socioeconomic status. This was exemplified by one HCP saying:

*I think that people who coming from a higher socioeconomic world, who grew up in some level that they take care more of their health than people that grew up with less money […]*.*(HCP 6)*

HCPs also described differences in comorbidities and genetic predisposition to certain diseases. One could argue that a higher prevalence of comorbidities affecting the self-care management of HF patients in certain cultures would create greater barriers to performing self-care for these populations. 

## 4. Discussion

This study provides insights into how healthcare providers experienced culture as a factor that influences or explains adherence to self-care advice for patients with chronic heart failure and how their own cultural background shapes their experiences and their practices. The findings illustrate that the cultural background of both the healthcare provider and the patient can affect the practice of providing self-care advice. It is therefore important that health care providers recognize one’s biases, prejudices, and assumptions about individuals who are different [19,22].

An important reflection of the healthcare providers was that culture permeates all facets of self-care and it is not enough to focus only on one behaviour or only one aspect of culture, e.g., food habits or religion. 

These findings also support differences found in other studies reflecting self-care differences in culturally diverse populations or in different countries worldwide [2,31,32]. Some types of self-care advice can interfere with strong cultural traditions or beliefs and the barriers may therefore be difficult to overcome, for example, culinary traditions, religion’s daily cooking, and attitudes to physical activity [33].

In this study, health care providers from different cultural backgrounds recognized the importance of self-care for their heart failure patients. However, from previous discussion in the literature it is important to consider that people with different backgrounds might have different goals for their self-care and the concept of self-care can be seen as being too focused on Western culture and might not been seen as important in cultures that are less focussed on autonomy. In some cultures individuals are less focussed on their own needs and autonomy as in many Western cultures but instead are more than focussed on not disrupting the happiness of others, even at the expense of their own quality of life [34].

Cultural traditions and beliefs or religion do not always support medical treatment or adherence to self-care [15,34,35]. In this study, it is was of particular interest that health care providers described their constant awareness of the cultural issues related to self-care and the impact it had on the care provided.

The cultural background of both the health care provider and the patient and the interplay between those were important considerations. A shared cultural background facilitated better understanding between health care provider and patient and the health care providers expected this to lead to stronger adherence to self-care advice in their patients. This is in line with findings from a US study in which black patients had highest adherence to medication when treated by a black healthcare provider [36]. Healthcare providers in our study described that they adapted their way of giving advice based on the cultural or religious background of their patients, but they were aware of issues of mistrust or prejudice. Such a mistrust or frustration because of non-optimal communication has been previously described [37]. Cultural distance is described as possibly negatively associated with the patients’ trust in their healthcare professionals and their ratings of quality of care [38]. 

Finally, as described in our findings, ‘culture is only part of the puzzle’; other factors might be more prominent than culture, for example, the different socioeconomic levels connected to a certain cultural group in certain circumstances [39].

In this study, we found that several different aspects of culture, economic situation and health knowledge were interrelated. The category ‘Mutual Cultural Background Impacts the Mindset to Address Self-Care’ described the complex matter of the culture of patients and health care professionals, and the professionals also described several issues that not only reflect the country people are born in or the religion people practice, but that sometimes the combination of several aspects coloured the perception of the health care professional. An intersectional theory approach is relevant for future studies in which micro- and macro-structures of power that operate in patients’ lives can be within the unique social spaces that they occupy. A recent study on stroke prevention in Arab immigrant women in Canada showed that there were unique pathways of privilege or vulnerability stemming from life stressors, lack of health literacy, and limited financial resources [40].

Summarizing, this study on the perspectives of health care providers gives important insight in the need to be aware of cultural background of both health care providers and patients. With increasing global migration, a wide cultural competence is needed. Earlier research has shown the benefits of specialised programmes aiming to improve cultural competence among healthcare providers, although more research is needed to provide evidence of better patient outcomes [20,25,41]. An education module for healthcare providers treating HF patients from different cultural backgrounds might be helpful for improving communication between healthcare providers and patients and thus possibly increasing patient adherence to self-care.

Finally, it is important to retain a patient-centred approach and, although adjusting care based on patients’ cultural background is vital, it is also important to consider that not all patients might identify themselves through their culture or feel a strong cultural identity.

## 5. Strengths and Limitations

This study has several strengths and limitations that need to be considered when interpreting the findings. The aim of the study was to gather information in a relatively unexplored area of research and provide a foundation for further research. To keep the discussion about culture broad, we did not define culture before the interview but instead ask the participants to reflect on what culture was for them first before starting the questions. We did not give them a definition on what we meant by culture but left it to the participant. On one hand, this can be seen as a strength, since we gave the participants the freedom to have a broad interpretation without steering them in a certain direction. On the other it might make the interpretation of the results less clear.

A strength of the study is that we included health care professionals from various cultural backgrounds who also treated patients with different cultural heritages. Talking about cultural differences is sensitive and carries the danger of talking about prejudice towards people from a certain culture. We acknowledged this in the interviews and ensured that the interview discussions were strictly confidential. The language differences might have created a barrier to fully understanding every participant, but most of the participants were fluent in English. The interviewer was not of Israeli origin and thus impartial, which might have facilitated unbiased discussions with regard to the different populations examined. Since there is a current political conflict in the country, this could have influenced the participants’ willingness to answer questions and might have coloured their answers.

Since we used a convenience sample and the main goal was not to focus on professions, we therefore did not actively look for an equal balance in health care professions. Future studies might look for a more balanced sample and even compare different professional background for similarities and differences. With the current data we could not compare perceptions between health care providers from different backgrounds and we recommend this for future studies.

## 6. Conclusions

In conclusion, the findings of this qualitative study suggest that clinicians perceive specific cultural and religious barriers to self-care for heart failure patients and that the provider’s own cultural background influences their care as well. Further research is necessary to confirm these suggested results and to assert the benefits of adjustments to care based on cultural background. 

## Figures and Tables

**Table 1 ijerph-17-05051-t001:** Demographic variables of participating health care providers.

Variable	Number of Participants
**Religion**	
Jewish	9
Muslim	2
Atheist	1
**Country of origin**	
Latvia	1
Russia	1
Israel	8
Great Britain	1
South Africa	1
**Gender**	
Male	10
Female	2
**Hospital**	
Rabin Medical Center	9
Soroka Hospital	3

**Table 2 ijerph-17-05051-t002:** Example of codes, subcategories and categories.

Code	Subcategory	Category
Differences in awareness of the importance of physical activity	Knowledge of health and education	Culture permeates self-care behaviours
Difference in understanding the importance of lifestyle changes		
It is not perceived as positive to stop smoking in certain cultures	Willingness to perform self-care	
Cultural differences in amount of medication taken		

**Table 3 ijerph-17-05051-t003:** Overview of the categories (bold) and subcategories in the total study.

**Culture permeates self-care behaviours**
Barriers to perform self-care, smoking, dietary changes or seeking help in case of deterioration Willingness to perform self-care Knowledge of health and education Habits and cultural norms Gender roles in different cultures
**Culture influences the way care is provided**
Adapting care and education seen as positive Adapting care and education seen as negative
**Mutual cultural background impacts the mindset to address self-care**
Shared background increases mutual understanding Different cultural backgrounds may lead to misunderstandings
**Culture is only a small piece of the puzzle…**
Belonging to a cultural group Other factors such as disease trajectory or socioeconomic status

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
