# Peer review of "Perspectives of Health Care Providers on the Role of Culture in the Self-Care of Patients with Chronic Heart Failure: A Qualitative Interview Study"

_ijerph, 2020, doi:10.3390/ijerph17145051_

Round 1

Reviewer 1 Report

This is a topic that would be of interest to readers (healt pssychologist in particular), the research design is good, and I appreciate the international context in which the study was conducted. However, precisely because the study is for an international audience it is absolutely necessary to broaden the review of the literature. I suggest to cite studies on Health Based Values perspective and also studies on health worker  and patient beliefs(ex. Steca et al., 2008; Capone, 2016; Campinha-Bacote, 2002).

I recommend that the authors provide more detailed description of the aims of the study rather than summary statements. It would make it easier for the reader to understand the justification for their methodological choices.

Participants and sampling. 

In my opinion, this paragraph is written in an unclear way. Rewrite it and better specify how the recruitment took place. 

Author Response

Thank you very much for your comments. Please see the attachment for our answers.

Reviewer 2 Report

This study describes the results of face-to-face semi-structured interviews on perceptions of caregivers on the influence of cultural background on self-care of patients with chronic heart failure.

Background:

The importance of self-care in heart disease and the influence of culture on its' adherence is well-described. I lack a rationale of why in depth qualitative analyses on perception of caregivers is neccessary and adding to the topic at this moment. 

Methods:

Design: Please provide more information on the type of interviews performed and the participants engaged.

Ethics: Did you have approval to collect the demographics of the participants?

Analysis: What software was used to code the interviews?

Results:

I lack a broad overview of the results of this study. I would like to see a table with an overview of topics and theme's identified, illustrated with quotes. It would be good to discuss barriers separately in the result section. 

It would also be very interesting to see which health care professionals stated which quotes. This can be done by stating the cultural background and eventually center of the persons mentioned in the quotes. This can be done between brackets. Were there any differences found in percenptions between the cultural backgrounds?

Discussion: Here you discuss the distinction of Jewish and Arab cultural groups, but this was not mentioned in the results. 

Did you ask participants for facilitators as well or solutions to improve the communication and adherence to self-care?

Author Response

thank you very much for your valuable comments, please see the attachment for our answers 

Round 2

Reviewer 1 Report

The manuscript has adequately improved.